# RNA and Single-Stranded DNA Phages: Unveiling the Promise from the Underexplored World of Viruses

**DOI:** 10.3390/ijms242317029

**Published:** 2023-12-01

**Authors:** Huong Minh Nguyen, Shinya Watanabe, Sultana Sharmin, Tomofumi Kawaguchi, Xin-Ee Tan, Dhammika Leshan Wannigama, Longzhu Cui

**Affiliations:** 1Division of Bacteriology, Department of Infection and Immunity, Jichi Medical University, Shimotsuke 329-0498, Tochigi, Japan; nguyen.mh@jichi.ac.jp (H.M.N.); swatanabe@jichi.ac.jp (S.W.); sultana.sharmin@jichi.ac.jp (S.S.); kwgctmfm@jichi.ac.jp (T.K.); xinee@jichi.ac.jp (X.-E.T.); 2Department of Infectious Diseases and Infection Control, Yamagata Prefectural Central Hospital, Yamagata 990-2292, Yamagata, Japan; leshanwannigama@gmail.com

**Keywords:** RNA phages, ssDNA phages, metaviromics, phage-based applications, genetic engineering, CRISPR/Cas-based genome editing, synthetic rebooting, phage therapy

## Abstract

RNA and single-stranded DNA (ssDNA) phages make up an understudied subset of bacteriophages that have been rapidly expanding in the last decade thanks to advancements in metaviromics. Since their discovery, applications of genetic engineering to ssDNA and RNA phages have revealed their immense potential for diverse applications in healthcare and biotechnology. In this review, we explore the past and present applications of this underexplored group of phages, particularly their current usage as therapeutic agents against multidrug-resistant bacteria. We also discuss engineering techniques such as recombinant expression, CRISPR/Cas-based genome editing, and synthetic rebooting of phage-like particles for their role in tailoring phages for disease treatment, imaging, biomaterial development, and delivery systems. Recent breakthroughs in RNA phage engineering techniques are especially highlighted. We conclude with a perspective on challenges and future prospects, emphasizing the untapped diversity of ssDNA and RNA phages and their potential to revolutionize biotechnology and medicine.

## 1. Introduction

First discovered independently by F. W. Twort in 1915 [1] and F. d’Herelle in 1917 [2] as a bacteria-eating virus, bacteriophages (or phages) are generally considered the most abundant biological entity on Earth, with an estimation of approximately 10^31^ particles in the biosphere [3]. Found in virtually all environments, phages exhibit unprecedented diverse morphology (polyhedral, filamentous, or tailed), genomic composition (DNA or RNA, double-stranded or single-stranded), and life cycle (lytic, lysogenic, or chronic productive infection) [4]. The use of phages to treat bacterial infections, called phage therapy, started in Eastern Europe from the early days of phage discovery but was later forgotten following the discovery of antibiotics [5]. The last few decades have seen antibiotic-resistant bacteria rise and persist as one of the major public health concerns worldwide [6], leading to renewed interest in the once-forgotten phage therapy. In recent years, treatments of bacterial infections using natural or engineered phages have restarted in Europe and America with notable successes and continue to gain momentum [7,8]. Together with antibacterial treatments, these triumphs have spurred phage research and engineering attempts to explore broader applications of phages in gene therapy, diagnostics and detection, drug delivery, cancer treatment, and vaccine development [9,10].

Until recently, most known phages were classified into one order, 14 families, and 37 genera, mainly based on their morphology [11]. With current advances in metagenomic and metatranscriptomic studies, however, in the most recent 2022 taxonomy update of the International Committee on Taxonomy of Viruses (ICTV), several major changes in phage classification have been officialized to put more focus on the genomic basis of these viruses. This includes the abolishment of the morphology-based families *Myoviridae*, *Podoviridae*, and *Siphoviridae*, and the replacement of the order *Caudovirales* by the class *Caudoviricetes* to include all tailed viruses of bacteria and archaea with icosahedral capsids and double-stranded DNA genomes [12]. Although the first single-stranded DNA (ssDNA), double-stranded RNA (dsRNA), and single-stranded RNA (ssRNA) phages were isolated back in the 1930s and 1970s [13,14,15], more than 90% of known phages to date are phages with double-stranded DNA (dsDNA) genomes [11]. By June 2019, only 12 representative ssRNA phage genome sequences were available from the NCBI Genome database [16]. A microvirus (ssDNA phage) dataset compiled in a study in 2022 included only 2147 genomes from GenBank; among these representative genome sequences of isolated virions were 542 [17]. Thus, it is unsurprising that dsDNA phages are the most thoroughly studied group and have played a pivotal, if not exclusive, role in many major phage-based discoveries and applications until now.

Despite accounting for less than 10% of all known phages to date, RNA and ssDNA phages possess multiple distinct features from dsDNA phages, such as their structure and genetic makeup, encoded lytic proteins, host range and receptors, or their life cycle, making them an attractive addition to dsDNA phages in various therapeutic applications. An exhaustive review of the hugely diverse RNA and ssDNA phages and their many practical applications is beyond the scope of this paper. In this review, we will attempt to deliver a concise account of the usages of this underexploited group of phages in various applications, focusing on their therapeutic usages as antimicrobial agents. To conclude, we will present our viewpoint on the future promise of engineering this group of phages for antimicrobial treatments, highlighting their potential advantages and distinctiveness.

## 2. A Brief History of ssDNA and RNA Phages

Although RNA and ssDNA phages were discovered not long after the isolation of the first dsDNA phage, this group has been historically understudied and underrepresented. Thanks to the recent rapid advancement of metagenomics and metatranscriptomics, the number of novel phage sequences, especially ssDNA and RNA phage sequences, has been significantly expanded. Substantial data on the evolution, hosts, life cycles, and molecular structure of ssRNA phages [18,19], dsRNA phages [20], and ssDNA phages [21,22] are available elsewhere. Here, we will briefly summarize relevant and current information about ssDNA and RNA phages to provide an overview of their important biological features and most recent taxonomical expansion (Table 1).

### 2.1. ssDNA Phages

The current classification separates ssDNA phages into five families with two distinct shapes: the tubular *Inoviridae*, *Paulinoviridae*, and *Plectroviridae*, and the icosahedral capsid *Microviridae* and *Finnlakeviridae*. These ssDNA phages have a circular genome of about 4.5–10.6 kb that encodes for 4–15 proteins; approximately half of these carry out morphogenesis and structural functions such as adhesion, replication, virion, and assembly. Besides these, unclassified ssDNA filamentous phage isolates (such as Vibrio phage K05K4, Genbank ID CP017905) or prophages [23] with genomes larger than 20 kb have also been identified. Except for a few well-studied ssDNA phages, such as M13 or phiX174, the functions of the remaining encoded proteins remain unknown [24,25]. Genomes of all ssDNA phages replicate through the rolling-circle mechanism with the involvement of host polymerase, converting the ssDNA into covalently bound dsDNA, known as the replicative form [24]. ssDNA phages are pili specific, binding to the tip of the host cell’s male pilus, such as Hfr, F+, and F’, via their coat protein (e.g., g3p in the case of Ff phages). The binding causes a conformational change of the coat protein, revealing the domain that can interact with another membrane-bound co-receptor of the host, TolA, facilitating phage genome entry [26].

Among the three families of tubular phages, *Inoviridae* and *Paulinoviridae* viral particles exist in a long, filamentous form, while *Plectroviridae* appears as a rigid rod-shape structure. As with all filamentous phages, members of these families lead a chronic productive infection life cycle where phage particles are released by extrusion without lysing their host cells. Interestingly, these three families infect completely different types of hosts. *Inoviridae* infect Gram-negative bacteria with a lipopolysaccharide (LPS)-containing outer membrane, while *Paulinoviridae* infect both Gram-negative and Gram-positive hosts and are found associated with mostly LPS-lacking Gram-negative hosts (ICTV-approved taxonomy proposals, *Paulinoviridae*, https://ictv.global/filebrowser/download/5770 (accessed on 26 November 2023)). *Plectroviridae* infect cell-wall-less bacteria, such as members of the *Mollicutes* order [27].

*Microviridae* and *Finnlakeviridae* comprise the other two families of ssDNA phages with small icosahedral capsids. The *Microviridae* family of ssDNA phages can be divided into two phylogenetic lineages, *Microvirus* and *Gokushovirinae*, based on the presence or absence of spike proteins on their capsids. *Microvirus* has 12 spikes on the 5-fold vertices, while *Gokushovirinae* lacks spikes but has mushroom-like protrusions at the 3-fold vertices [22]. Initially, all cultured phages of the *Microviridae* family were isolated from *Enterobacteriaceae* (in the case of *Microvirus*) and intracellular parasites (in the case of *Gokushovirinae*), leading to the belief that this group of phages has a narrow host range. The culturing technique, however, can lead to underestimation, as prophages of gokushoviruses were later found in *Bacteroidetes* genomes using the prophage detection method [28]. Recent metagenomics surveys revealed that microviruses are much more prevalent and present in most environments, highlighting their ecological importance [29].

*Finnlakeviridae* is a recently recognized ssDNA phage family that includes a single genus, *Finnlakevirus*, with only one species, *Finnlakevirus FLiP*, isolated from a boreal freshwater habitat in Central Finland in 2010 [30]. It is the first described ssDNA phage that contains an internal membrane enclosed in the icosahedral capsid [31]. The striking similarity between the capsid proteins of *Finnlakevirus FLiP* and dsDNA viruses of the PRD1-adenovirus lineage suggested a possible evolutionary relatedness between some ssDNA and dsDNA viruses.

**Table 1 ijms-24-17029-t001:** Current classification of RNA and ssDNA phages.

Group	Family	Virion	Genome	Replication	Host Range	Life Cycle	Members ^1^	Example [References]
ssDNA	*Inoviridae*	Non-enveloped flexible filaments	Circular +ssDNA	Rolling circle	LPS^+^ Gram-negative bacteria	Chronic infection	25 genera, 43 species	Phage M13 [24]
	*Paulinoviridae*	Non-enveloped flexible filaments	Circular +ssDNA	Rolling circle	LPS^−^ Gram-negative and -positive bacteria	Chronic infection	2 genera, 2 species	Phage B5; phage OH3 [27,32]
	*Plectroviridae*	Non-enveloped rigid rods	Circular +ssDNA	Rolling circle or transposition	Cell wall-less bacteria	Chronic infection	4 genera, 6 species	Phage MV-L1 [33]
	*Microviridae*	Non-enveloped icosahedral virions, spikes −/+	Circular +ssDNA	Rolling circle and other mechanism(s)	*Enterobacteria*, intracellular parasitic bacteria, cell wall-less bacteria	Lytic	7 genera, 22 species	Phage φX174; phage 4 [34]
	*Finnlakeviridae*	Icosahedral virion with spikes, internal lipid membrane	Circular +ssDNA	Possibly rolling circle	Gram-negative *Flavobacterium*	Lytic	1 genus, 1 species	Phage FLiP [30]
dsRNA	*Cystoviridae*	Enveloped multi-layer icosahedral virions with spikes	Segmented, linear dsRNA	ssRNA → dsRNA	Gram-negative bacteria, mostly *Pseudomonas*	Lytic	1 genus, 7 species	Phage phi6 [35]
ssRNA (before 2021)	*Leviviridae*	Non-enveloped icosahedral virions	Linear +ssRNA	−ssRNA → +ssRNA	Gram-negative bacteria	Lytic	2 genera, 4 species	Phage MS2 [36], phage Qβ [34]
ssRNA (since 2021) ^3^	*Atkinsviridae*	NA ^2^	NA ^2^	NA ^2^	NA ^2^	NA ^2^	56 genera, 91 species	Uncultured viral genomes [37]
	*Duinviridae*	NA ^2^	NA ^2^	NA ^2^	NA ^2^	NA ^2^	6 genera, 6 species	Uncultured viral genomes [37]
	*Fiersviridae * (formerly *Leviviridae*)	Non-enveloped icosahedral virions	Linear +ssRNA	−ssRNA → +ssRNA	Gram-negative bacteria	Lytic	185 genera, 298 species	Phage MS2 [36], phage Qβ [37]
	*Solspiviridae*	NA ^2^	NA ^2^	NA ^2^	NA ^2^	NA ^2^	24 genera, 31 species	Uncultured viral genomes [37]
	*Blumeviridae*	NA ^2^	NA ^2^	NA ^2^	NA ^2^	NA ^2^	31 genera, 35 species	Uncultured viral genomes [37]
	*Steitzviridae*	NA ^2^	NA ^2^	NA ^2^	NA ^2^	NA ^2^	117 genera, 412 species	Uncultured viral genomes [37]
	Unassigned	NA ^2^	NA ^2^	NA ^2^	NA ^2^	NA ^2^	9 genera, 9 species	Uncultured viral genomes [37]

^1^ https://ictv.global/msl (ICTV_Master_Species_List_2022_MSL38.v3.xlsx, created 9 November 2023—11:17) (accessed on 26 November 2023). ^2^ NA: not available. ^3^ Due to the hyperextension of discovered ssRNA phage sequences from metatranscriptomes, since 2021, this group includes 882 species and their taxonomy has been completely restructured. Noted that many species are sequence-only or uncultured viral genomes.

### 2.2. dsRNA Phages

RNA (dsRNA and ssRNA) phages constitute the smallest portion of known bacteriophages. Currently, ICTV registers seven species of dsRNA phages, all belonging to the single family *Cystoviridae*. The family *Cystoviridae* includes enveloped viruses with a tri-segmented dsRNA genome enclosed in one or two concentric, icosahedrally symmetric protein shells. The innermost protein layer is a polymerase complex responsible for genome packaging, replication, and transcription [35].

All known dsRNA phages infect Gram-negative bacteria, mainly plant-pathogenic *Pseudomonas syringae* strains [35]. Until 1999, the *Pseudomonas* phage phi6 was the sole species of the *Cystoviridae* family [38]. Additional lipid-containing, three-segmented dsRNA phages were subsequently isolated into two groups: Those closely related (phi 7, phi9, phi10, and phi11) and those distantly related (phi8 and phi12) to phi6. Phi13 lies between the two groups [39]. Phi6 and its closely related relatives utilize type IV pili as the host receptor, limiting their infection range to *P. syringae* and several mutants of *P. pseudoalcaligenes* ERA [39]. In contrast, phi8 and phi12 infect *P. syringae* with rough LPS [38]. Phi8, phi12, and phi13 could infect other rough-LPS-containing Gram-negative bacteria, though plaque formation was not observed when phi12 was used to infect JM109 [39]. Two other *P. syringae* dsRNA phages, phi2954 and phiNN, were later isolated from radish leaves [40] and freshwater habitat [41], respectively; both utilize type IV pili as the host receptor. In 2016, the first dsRNA phage infecting the human pathogen *P. aeruginosa* was successfully isolated from hospital sewage in China [42]. These studies demonstrate a wider distribution and broader host range than previously anticipated for these dsRNA viruses.

### 2.3. ssRNA Phages

In previous ICTV taxonomy releases, the ssRNA phages were grouped into a single family, the *Leviviridae*, consisting of two genera and four species: *Levivirus* with two species (MS2 and BZ13) and *Allolevivirus* with two species (Qβ and F1) [34]. All ssRNA phages have small icosahedral virion encapsulating a linear, plus-strand RNA genome that encodes for four proteins: the coat protein, the replicase, the maturation protein, and the lysis protein (read-through protein in the case of *Allolevirus)* [43]. Similar to ssDNA phages, RNA phages also utilize the host’s pili as their binding site, although instead of binding to the tip of the pilus, both dsRNA and ssRNA phages bind to the side of the pilus to start their infection cycle [44,45,46]. ssRNA phages bind to their hosts via its maturation protein [47], which also interacts with the 3′ end of the genomic RNA and enters the host cell together with the genomic RNA following adsorption [48].

By 2019, only 12 genome sequences of ssRNA phages were available from the NCBI database, leading to the assumption that this group of phages is scarce. Recent metagenomic and metatranscriptomic studies, however, have revealed that ssRNA phages are much more diverse and abundant than previously thought, expanding the number of known genomes from this group from tens to over a thousand [16], resulting in their complete taxonomical restructuring [37]. ssRNA phage sequences were detected on three continents, showing their widespread global distribution. ssRNA phages are now ratified into the *Leviviricetes* class, encompassing two orders (*Norzivirales* and *Timlovirales*) with six families, 428 genera, and 882 species (https://ictv.global/msl (ICTV_Master_Species_List_2022_MSL38.v3.xlsx, created 9 November 2023—11:17) (accessed on 26 November 2023)). It is safe to say that RNA phages, including both dsRNA and ssRNA phages, are the fastest-growing group of phages nowadays.

### 2.4. Current Opportunities and Challenges in Metaviromic Studies

Over the last decades, enabled by high-throughput sequencing technologies, an unprecedented diversity of microbial and viral communities has been revealed from various environmental and biological samples [16,17,29,49,50,51,52]. For a long time, however, sample preparation and analysis methods for metagenomic studies were mainly developed and optimized with a focus on dsDNA. This led to a significant underpresentation of ssDNA samples. Similarly, metatranscriptomic studies of the RNA virome also face various challenges concerning sample preparation and analysis. Most metaviromic studies proceed through the basic steps of isolating and concentrating viral samples, followed by nucleic acid extraction and library preparation. Without careful consideration, biases can be introduced in each step [53]. Szekely and colleagues [22] listed several potential biases in sample preparation due to the fundamental differences between dsDNA and ssDNA phages, identifying the two most prominent biases that could lead to underpresentation of ssDNA phages: Different buoyant density and library amplification methods. The authors recommended recovering viral samples at a lower buoyant density and preparing the metagenomic library using multiple displacement amplification (MDA) method to ensure better enrichment of ssDNA samples. There is also promise in the use of library preparation kits that employ ssDNA as starting material to overcome biases from the MDA method.

Following nucleic acid extraction, metatranscriptomic samples often proceed with ribosomal RNA (rRNA) removal and then complementary DNA (cDNA) generation by reverse transcription. Subsequently, random primers are used to enrich the resultant cDNA library, followed by adapter ligation and sequencing [54]. rRNA-reduced libraries are shown to produce more viral contigs than polyadenylation-selected libraries [55], proving their suitability for RNA phage samples. The problem of low abundancy and incomplete dsRNA viral genome retrieval in metatranscriptomes was addressed by Urayama and colleagues using their new sequencing method named fragmented and primer-ligated dsRNA sequencing (FLDS) [56]. The remaining major bottleneck in metatranscriptome analysis is the lack of reference sequences in less well-studied viral groups, such as bacterial RNA viruses. To this end, the strategy employed by Neri and colleagues combined RNA-dependent RNA polymerase (RdRP)-based discovery, matching bacterial and viral CRISPR spacers, and identification of bacteriolytic proteins was shown to be effective [52]. Despite multiple challenges, the field of metaviromic studies is expected to continue expanding rapidly, unveiling broader and deeper knowledge of the virosphere, including the ssDNA and RNA phageome.

## 3. Genetic Engineering of RNA and ssDNA Phages

RNA and ssDNA phages have been pivotal models in molecular biology since the early days, playing a significant role in understanding genetic code, RNA translation, and virus-host interactions [57]. These studies have paved the way for the development of multiple practical applications such as bio-imaging [58], affinity purification [59], bacterial detection and control [60], gene editing [61], gene and drug delivery, vaccine development, and cancer treatment [62]. Most recently, the use of ssDNA and RNA phage-derived components in synthetic biology, a younger branch of modern biotechnology that arose roughly two decades ago, has led to the development of various useful tools such as DNA origami [63], riboswitches [64], and multiple other therapeutic RNA modalities. In this section, we highlight examples of the major contributions of ssDNA and RNA phages to biotechnology (Table 2).

In addition to providing tools and technologies for modern biology, ssDNA and RNA phages have also been employed to control bacteria and treat infections. Recent developments in phage engineering bring novel therapeutic values to phage therapy, creating new prospects for its applications in medical treatments. Our group, among others, has pioneered the development of phage-based RNA-cleaving antibacterials, encompassing sequence-specific bacterial gene detection, selective elimination of drug-resistant bacteria, and targeted manipulation of the bacterial flora [65]. Although not yet as widely utilized as dsDNA phages, ssDNA and RNA phages also make substantial contributions to antimicrobial therapy, a topic we will delve into in the next section, along with our related research results.

**Table 2 ijms-24-17029-t002:** Tools and technologies based on genetic engineering of ssDNA and RNA phages.

Phage Components ^1^	Tools and Technologies	Applications ^1^	Reference(s)
ssDNA phages	Phage display	Library screening	[66,67,68,69]
		Cancer treatment	[70,71,72]
	Cell adhesive substrates using electrospinning	Drug delivery	[73,74,75]
	Carbon nanotubes	Biosensing and imaging	[76,77,78,79,80,81]
dsRNA phages	dsRNA production	RNAi-based crop protection	[82]
	Surrogate model	dsRNA virus research	[83,84,85]
CP and TR of ssRNA phages	Protein–RNA tethering		
	Single tagging	In vivo RNA imaging	[58,86]
		Riboswitch screening	[64]
	Dual tagging	In vivo two-color RNA imaging	[87,88,89]
	Affinity purification		
	RNA–protein	In vitro complex	[59]
		In vivo complex	[90]
	RNA–RNA	In vivo complex	[91]
	CRISPR/Cas9-based gene regulation	In vivo transcription activation	[61,92]
Recombinant VLPs of ssRNA phages	Therapeutic display		
	Peptide	Cell targeting and penetrating	[93,94]
	Glycan	Cell targeting	[95]
	DNA aptamer	Cell targeting	[96]
	Antibody	Cell targeting	[97]
	Therapeutic delivery		
	RNAs	Small interfering RNA delivery	[98]
	Toxins	Protein toxin delivery	[99]
	Small molecules	Chemotherapeutic delivery	[100]
	Antigen display	Vaccine development	[101,102]
	RNA cargo	RNA vaccine development	[103]
	Armored RNA	RNA virus detection	[104,105]
	Nanoreactor	Controlled enzymatic reaction	[106,107]

^1^ CP: coat protein, TR: translation repression, VLP: virus-like particles, RNAi: RNA interference.

### 3.1. Engineering ssDNA Phages

Among ssDNA phages, filamentous ssDNA M13 phage is commonly employed for practical and therapeutic applications, utilizing engineering platforms such as genetic engineering for phage display, electrospinning for bionanomaterial development, and phage-directed nanomaterial combinations [75]. Phage display, a genetic engineering process, involves the insertion of foreign peptide coding sequences into phage capsid genes, resulting in the display of corresponding peptides on the phage surface [66,67,68]. Libraries of displayed peptides are often constructed, followed by rounds of affinity selection (called biopanning) to select the most suitable therapeutic candidates [69]. Using M13 phage, phage display technology has been applied to identify cancer cell surface markers, leading to the development of functional anticancer peptides for therapeutic treatment [70]. In this study, Wang and colleagues used M13 phages to display the antitumor cytokine GM-CSF, a potent activator of STAT5 signaling in macrophages of rodents, to target colorectal cancer cells. Post-GM-CSF M13 administration, the tumor size was significantly attenuated, and the number of CD4+ lymphocytes increased. Combining GM-CSF M13 phage therapy and radiation resulted in a 100% survival rate and a 25% complete mitigation rate in mice. In another study, Jin and colleagues applied engineered M13 phage to selectively bind to different types of collagens using collagen-mimetic peptide motifs. The engineered M13 phages could target and label abnormal collagens, which were then easily detected by fluorescence imaging, enabling monitoring and diagnosis of various pathological conditions [72]. Furthermore, in the case of cardiovascular disease, Lee and colleagues developed an M13 phage-based double functional peptide-carrying system where RGD peptides (a cell adhesive motif) were displayed in the pIII minor coat proteins to bind with integrin-expressing cells for constructing an artificial niche [108]. The engineered M13 nanofiber dramatically enhanced ischemic neovascularization by activating intracellular and extracellular processes such as proliferation, migration, and tube formation in the endothelial progenitor cells (EPCs) after transplanting M13-phage-treated EPCs into a mouse hindlimb ischemia model.

“Electrospinning” is the construction technique of fibers ranging from nanometers to micrometers by passing polymer solutions through a highly electrically charged environment to generate fibrous membranes. Shin and colleagues used electrospinning to create hybrid nanofiber sheets from a mixture of poly-lactic-co-glycolic acid (PLGA) and peptide-displaying M13 phages, generating an ideal cell-adhesive substrate [73]. These nanofibers can be used as efficient drug delivery systems for various therapies, incorporating bioactive molecules such as antibiotics, anti-inflammatories, anti-cancer drugs, genomic DNA, proteins, probiotics, and enzymes [74,75].

Phage-directed nanomaterial combinations involve compounds’ genetic and chemical integration to create phage-directed fusion substances for biosensing and scaffold building [79]. Carbon nanotubes have been assembled on M13 phage surfaces for detecting bacteria and tumors through fluorescence imaging [76,77]. Researchers constructed M13-SWNT (single-walled carbon nanotube) conjugates and demonstrated comparable effects on monitoring specific bacteria, prostate cancer, and ovarian cancer, respectively [76,77,78]. Combining M13 phage with metal nanoparticles has also been demonstrated to improve fluorescence bioimaging for detecting bacteria [80] and cancer cells [81]. Dong and colleagues used silver nanoparticle-conjugated M13 phages that caused deconstruction of bacterial cell walls and intracellular biomolecules, inducing oxidative stress that killed pro-tumoral bacteria for gut microbiota regulation [71]. Thus, engineered M13 phage has versatile applications in different aspects of bionanomaterial that is useful in drug delivery, biodetection, tissue regeneration, and targeted cancer therapy.

### 3.2. Engineering dsRNA Phages

Phi6 has found utility in the production of antiviral dsRNA for crop protection [82]. Loss by crop pathogens and pests is estimated at around USD 100 billion annually, and resistance against chemical pesticides is rising. A promising alternative approach is the application of double-stranded RNA (dsRNA), which triggers RNA interference (RNAi), an antiviral defense mechanism in eukaryotes that cleaves invading viral RNA genomes and represses translation of viral transcripts [109]. Previously, antiviral dsRNAs were produced through post-synthetic hybridization of in vitro or in vivo transcribed ssRNAs from DNA templates, but this approach is costly and inefficient. By using the recombinant protein P2 of phi6, Makeyev and colleagues could efficiently produce double-stranded RNA in vitro from positive-sense RNA substrates [110]. Niehl and colleagues later developed an in vivo dsRNA production system in *P. syringae* using constructs carrying the phi6 replication complex and different targets inserted into the phi6 genomic S (small) and M (medium) segments, observing significant dsRNA-mediated inhibition of tobacco mosaic virus (TMV) propagation in the tested disease model [82]. The established platform thus provides an economical and efficient large-scale production of multiple long dsRNAs for various therapeutic applications. Future research may uncover additional effective methods for producing dsRNA from the replication machinery of other dsRNA phages, a topic that has not yet been thoroughly explored.

dsRNA phages have also been used as a surrogate model in studies of human pathogenic RNA viruses such as coronavirus, SARS-CoV-2, influenza, and Ebola [83,84,85] due to their similar size and structural organization. The non-enveloped ssRNA phage MS2 and the ssDNA phage PhiX174 often served as additional controls in these studies. With their segmented dsRNA genome, dsRNA phages share similarities with the family Reoviridae, a diverse family consisting of plant, fungi, invertebrate, and vertebrate viruses. Therefore, dsRNA phages provide opportunities to study the genome packaging and assembly mechanisms of dsRNA-segmented genome viruses in a simple and easy-to-handle prokaryotic system [38].

### 3.3. Engineering ssRNA Phages

Prominent applications of ssRNA phage involve various protein-RNA tethering systems and recombinant virus-like particles (VLPs) (see [111] for a detailed review). Protein-RNA tethering systems were established by exploiting the distinctive ability of different ssRNA phage coat proteins (CP) to recognize and bind to their specific cognate RNA stem loop operators known as TR (translation repression) [112,113,114]. An efficient in vivo imaging system for mRNA was developed by Tyagi, utilizing reporter-fused CPs and TR-tagged RNAs [86]. This methodology has since been proven effective in tracking the processing, import and export, localization, translation, and degradation of tagged mRNA [58]. Wu and colleagues conducted a high-throughput experiment on 2875 sensor molecules by utilizing the binding of fluorescently tagged MS2 CPs to the MS2 TR segment within designed riboswitches. This approach, combined with the recently developed RNA on a massively parallel array (RNA-MaP) platform, allowed for the testing of diverse riboswitch designs [64]. Additionally, dual tagging systems have been devised, employing either fluorescent protein fragments to reduce background signal [87,88] or two different fluorescent proteins to enable two-colored labeling of a single-molecule RNA [89].

The CP–TR interaction was also employed to develop affinity purification methods for RNA–protein [59] and RNA–RNA [91] complexes. Bardwell and colleagues utilized resin-bound R17 CP to capture tandem-TR-tagged mRNA along with its binding factors. Tsai and colleagues [90] implemented MS2 BioTRAP, incorporating histidine and biotin (HB)-tagged MS2 CPs co-expressed with TR-tagged RNA targets, to capture and quantify in vivo interactions between target RNA and protein factors. A similar approach was used to identify miRNA interactions with their target mRNAs [91], thus expanding the application of CP as an in vivo RNA–RNA tether molecule.

The MS2 CP–TR tethering system has found application in CRISPR/Cas9 technology for orthogonal gene knockout and transcriptional activation in human cells [61]. Dahlman and colleagues achieved this by incorporating the MS2 TR loops into the shortened sgRNA, effectively inhibiting Cas9′s nuclease activity while preserving its binding to the target promoter sequence. This strategic manipulation facilitated recruitment of the transcription activator VP64 through MS2 TR loops, resulting in potent upregulation of gene expression. Beyond its role in simultaneous gene knockout and transcriptional activation, this platform has also been employed for genome-scale gain-of-function screening, particularly in studying drug resistance in a melanoma model [92].

Recombinant VLPs of ssRNA phages can be generated by expressing the CP gene in bacteria or yeast [115,116]. Due to their single protein composition, these VLPs offer a more straightforward engineering process compared to most VLPs derived from other phages and viruses. They can be decorated with peptides [93,94], glycans [95], DNA aptamers [96], or antibodies [97] with cell-targeting and cell-penetrating capabilities. These decorated VLPs, in turn, serve as carriers for therapeutic agents such as RNAs [98], toxins [99], nanoparticles, or small molecules [100] directed towards desired targets. In the realm of vaccine development, VLPs play a dual role—displaying immunogenic components on their surface [101] or carrying them internally [103]. Moreover, VLPs themselves contribute as adjuvants thanks to their potent immunogenic properties (for an up-to-date review of RNA phage VLP-based vaccine platforms, refer to [102]).

Two other noteworthy applications employing ssRNA phage VLPs are armored RNA [104] and nanoreactors [106,107]. In the first application, VLPs serve as protective cages, safeguarding control RNAs in various RT-qPCR-based RNA virus detection systems from unwanted RNase degradation [105]. In the second application, VLPs act as carriers for encapsulating functional enzymes, facilitating controlled biochemical reactions [107]. The commonly used strategy for packaging therapeutic cargoes within ssRNA phage VLPs involves the conjugation with the TR RNA stem loop, achieved either through genetic engineering in the case of RNA cargoes [104] or through chemical conjugation methods for cargoes of different natures [106].

### 3.4. Techniques for Genetic Engineering of ssDNA and RNA Phages

The continuous evolution of technology to manipulate phages for addressing healthcare and economic challenges is expected. Engineered phages hold promise for carrying exogenous DNA or RNA, as well as displaying functional molecules for application in disease treatment and prevention, imaging and detection, biomaterial development, and diverse delivery systems. In this section, we will briefly outline the fundamental techniques commonly employed in phage engineering. These include recombinant expression of phage-derived products, engineering phage genomes through homologous and non-homologous recombination, CRISPR/Cas-based phage genome editing, and synthetic rebooting of phage-like particles. Additionally, we will delve into recently developed techniques for engineering RNA phages, as illustrated in Figure 1.

Recombinant expression of phage proteins is typically conducted in natural host cells or closely related counterparts. Unlike dsDNA phages, ssRNA phage VLPs exhibit self-assemble capabilities when recombinant coat proteins are present alongside suitable genomic DNA or RNA fragments. Therefore, a dual-plasmid system is commonly employed for co-expressing the phage’s coat protein and copies of relevant viral genomic segments. This approach facilitates the generation of diverse functional phage particles, allowing the insertion of desired cargoes within the genomic fragment or their display on the virion surface (for a comprehensive review, refer to [117]).

Thanks to the small genome size, engineering of ssDNA and RNA phage genomes can be performed through standard molecular cloning and plasmid transformation techniques into host cells. The reverse genetic system for RNA viruses uses complementary DNA (cDNA) to introduce desired mutations into the RNA genomes [118]. Modifications of cDNA, serving as templates for phage protein synthesis and RNA replication, in turn allow for studies of the viral life cycle and engineering of these viruses for practical applications [119,120]. For dsRNA phages such as phi6 and phi8, a carrier state can be established by electroporating non-replicative plasmids containing cDNA copies of viral genomes into host cells [121]. This construction allows for the incorporation of marker genes such as *kan* or *lacα* inside the phage virions. Phagemids, carrying phage origin of replication and packaging sites, are efficiently replicated and packaged by suitable phage machinery provided in *trans* from a separate helper phage [122]. Phagemids are widely employed in generating both dsDNA and ssDNA phage particles, encapsulating and displaying exogenous therapeutic elements [123,124].

**Figure 1 ijms-24-17029-f001:**
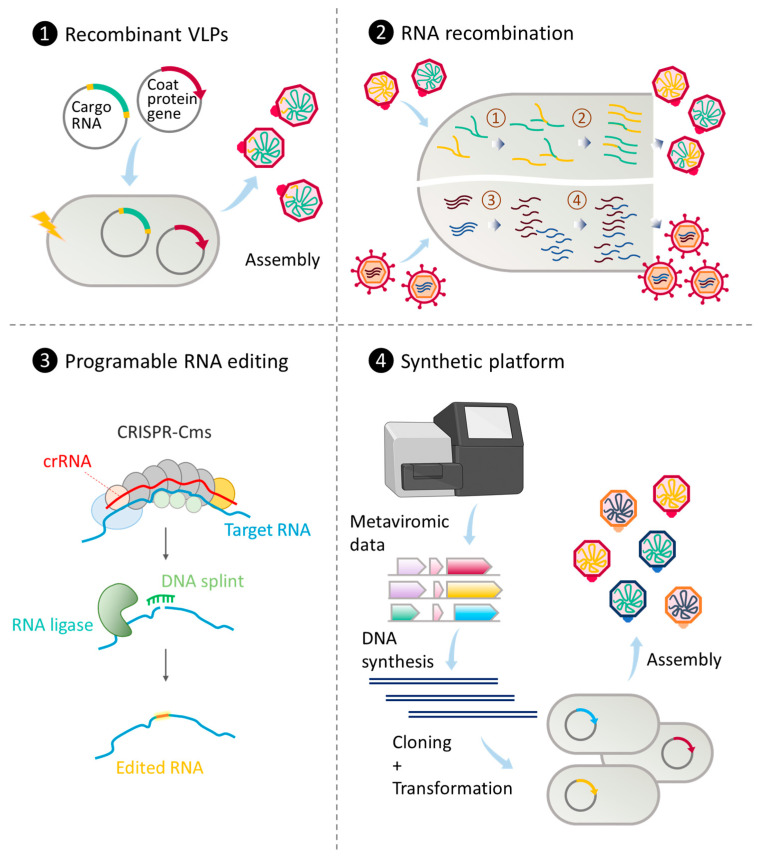
Strategies to engineer RNA phages for practical and therapeutic purposes. (1) Co-expression of phage coat proteins and suitable RNA templates from transformed plasmids. Recombinant VLPs can be assembled through the direct interaction between coat proteins and RNA molecules [125]. (2) RNA recombination through co-infection. When two ssRNA genomes replicate (step ①), template switching can occur, leading to the production of hybrid genomes (step ②) and subsequently, hybrid phages [126]. A similar recombination phenomenon can occur during dsRNA phage replication (step ③), after which genome reassortment can lead to various combinations of packaged genomic segments (step ④) [127]. (3) Programmable RNA editing using guide RNA (crRNA) and CRISPR-guided nuclease (CRISPR-Cms) cleavage followed by RNA ligase-mediated strand repair in the presence of a synthetic DNA splint (adapted from [128]). (4) Synthetic platform for the production of viral VLPs from metaviromic data. Predicted genes encoding coat proteins are chemically synthesized, cloned, and transformed into bacterial hosts. Successful expression will lead to the assembly and release of virus-like particles [129].

Natural recombination, whether homologous or heterologous, is frequently observed in positive-sense single-stranded RNA viruses, including ssRNA phages. Viable hybrid progenies between different RNA phage species can be generated through a two-plasmid co-transformation system into bacterial hosts [130]. In vitro recombination of non-replicating genomic RNA fragments with overlapping ends in the presence of Qβ replicase has been observed, resulting in replicative Qβ genomes [131]. However, it is important to note that recombination in RNA viruses often yields multiple outcomes, reducing the precision of modification. A technique known as CRE-REP, established by Lowry and colleagues, utilizes well-defined non-viable “donor” and “receptor” RNA genomic fragments to enable the selection of viable recombinants from cell-based recombination upon co-transfection. This approach has significantly improved the detection and quantification of recombination events [132]. Originally developed for poliovirus, the technique has been adapted for use with other viruses [133,134].

In September 2023, Nemudryi and colleagues unveiled an innovative CRISPR/Cas-based RNA editing technique facilitating rapid and programmable deletions, insertions, and substitutions in RNA without the need for DNA intermediates [128]. By combining type III CRISPR/Cas-based RNA cleavage with splinted RNA ligation, the authors achieved targeted modifications in RNA sequences. This method, free from DNA intermediates, offers a workaround for the reliance on known hosts, a prevalent bottleneck in the study and engineering of most RNA phages. This platform holds great potential for robust engineering of a wide range of RNA viruses with precision and efficiency.

Recently, chemical synthesis of novel phage sequences or genomes from metavirome data, followed by cloning and heterologous expression, or rebooting, has become a powerful technique to study and engineer the richly diverse novel RNA phages without extensive prior knowledge. For instance, Lieknina and colleagues synthesized, cloned, and overexpressed 110 coat protein genes from 150 novel, uncultured ssRNA phage sequences identified from metagenomic data. This effort successfully yielded 80 assembled VLPs [129]. The obtained VLPs exhibited variations in size, shape, stability, and assembly temperature, with some demonstrating specific interactions with different candidate RNA structures. This synthetic platform can be applied to virtually all annotatable protein sequences of RNA phages retrieved from the metaviromic data. Furthermore, rebooted dsRNA phages from synthetic cDNA clones have been used to broaden the host range of the studied phages [135], presenting an additional avenue for engineering RNA phages with flexibility and robustness for diverse purposes.

## 4. Current Applications of RNA and ssDNA Phages as Therapeutic Agents against Multi-Drug Resistant Bacteria

Antibiotics play a crucial role in treating bacterial infections, but their overuse and misuse have led to the emergence of antimicrobial resistance (AMR), resulting in reduced drug efficiency and persistent infections. Given the rapid evolution of AMR and the challenges of developing novel drugs, exploring alternative strategies becomes imperative. Phage therapy, utilizing phage-derived products and both natural and engineered phages for infection treatment, along with infection prevention through vaccines targeting antimicrobial-resistant pathogens, stands out as a promising frontier in the battle against AMR [136]. In the following sections, we discuss the contribution of ssDNA and RNA phages as antibacterial therapeutics, highlighting their advantages and unique features (see Figure 2).

### 4.1. Phage-Derived Lytic Enzymes as Antibacterial Agents

Phage-derived lysins, also known as “enzybiotics,” present a promising alternative to conventional antibiotics. These enzymes exert their antibacterial effects by lysing host bacteria through the cleavage of peptidoglycan (PG), the primary structural component of the bacterial cell wall. Phage-derived lytic enzymes fall into two categories: virion-associated peptidoglycan hydrolases (VAPGHs) and endolysins. VAPGHs initiate degradation of PG at the onset of the phage infection [137,138], whereas endolysins act at the conclusion of the phage infection cycle, facilitating the release of mature phages [139]. In most cases, dsDNA phages encode distinct VAPGHs and endolysins, whereas dsRNA phages feature a single lytic protein that serves as both VAPGH and endolysin [140]. In contrast, ssDNA and ssRNA phages employ a single-gene lysis (Sgl) protein—an impactful degradative protein that induces cytolysis without enzymatically breaking down PGs [141,142].

**Figure 2 ijms-24-17029-f002:**
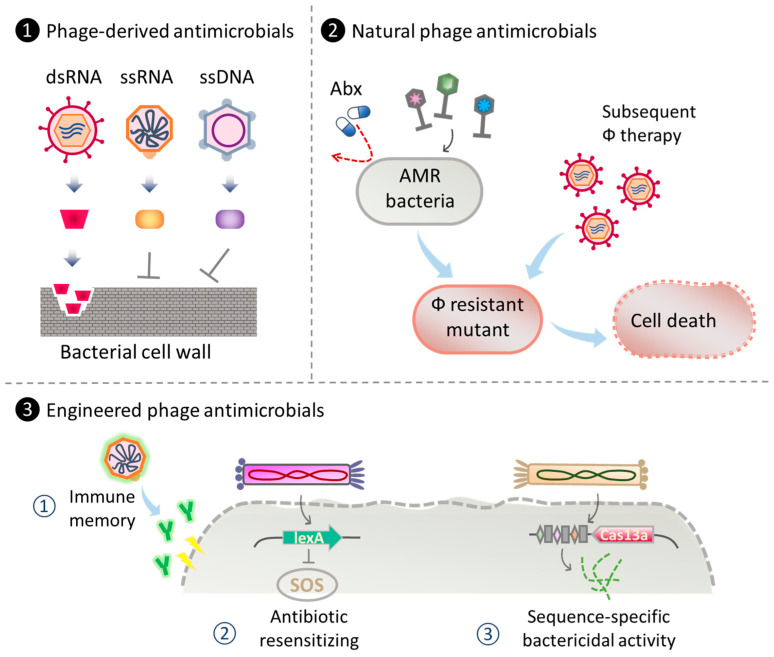
Strategies for phage therapy using ssDNA and RNA phages and their products. (1) Utilization of phage-derived components as antibacterial agents. For instance, the degradation of bacterial cell wall by dsRNA phage lytic enzyme [143] or inhibition of cell wall synthesis using single-gene lysis (Slg) protein from ssDNA and ssRNA phages [141,142,144]. (2) Phage cocktail employing natural ssDNA and RNA phages as antimicrobial therapeutics against AMR bacteria and phage-resistant mutants [145,146]. (3) Engineering ssDNA and RNA phages for antibacterial therapies. Examples include ① engineered phages displaying specific antigens as vaccine against drug-resistant bacteria [147,148]; ② engineered phages to enhance antibiotic efficacy in phage-antibiotic combinatory therapy [149]; ③ engineered antibacterial capsids carrying CRISPR/Cas13a for sequence-specific-based bacterial gene detection, bacterial flora modification, and treatment for AMR bacterial infections [65]. Abx: antibiotic.

Ply17, a lytic enzyme produced by *Pseudomonas* dsRNA phage phiYY, has demonstrated the capability to reduce the number of viable Gram-negative bacteria such as *P. aeruginosa* and *E. coli*, as well as Gram-positive bacteria including *S. aureus* and *S. epidermidis,* by approximately 2 logs when treated with an outer membrane permeabilizer such as EDTA [143]. Notably, recent genome analyses of the animal dsRNA viruses from the *Picobirnaviridae* and *Partitiviridae* families have unveiled potential antibacterial and antifungal lysis genes [52]. Upon cloning and expression of the lytic genes from *Picobirnavirus* and *Partitivirus* in *E. coli* DH5a, the observed growth inhibition was comparable to that induced by the *Enterobacteria* MS2 phage lysin L [150].

The bacteriolytic activity of ssRNA *Leviviridae* phages varies among different phages. For example, Sgl_M_ of phiM and Sgl_PP7_ of *Pseudomonas* phage PP7 exhibit bacteriolytic activity by inhibiting MurJ, the transporter responsible for moving lipid-binding peptidoglycan precursors from the inside to the outside of the plasma membrane [141,142]. On the other hand, Sgl_Qβ_ from Qβ phage exhibits bacteriolytic activity by non-competitively inhibiting MurA, the first enzyme in the PG biosynthetic pathway [151]. MS2 Sgl_MS2_ (lysin L) possesses an N-terminal heat shock-responsive chaperone, DnaJ, but its specific mechanism of action remains unclear. Other known Sgls include Sgl_PRR1_ (*Pseudomonas* phage PRR1), Sgl_KU1_ (Enterobacteria phage KU1), and Sgl_Hgal1_ (Enterobacteria phage Hgal1), whose mechanisms of action have not yet been elucidated [142].

Sgl_φ174X_, a bacteriolytic enzyme produced by the ssDNA phage phiX174, inhibits MraY, an enzyme catalyzing the initial step of the lipid cycle reaction in PG biosynthesis. This inhibition results in bacteriolysis [142,144]. As the cellular targets and modes of action of many ssRNA and ssDNA phage Slgs are still unidentified, future genome analyses hold great capacity for the discovery of new therapeutic candidates.

### 4.2. Natural ssDNA and RNA Phages as Antibacterial Agents

*Pseudomonas* phages phiYY [42] and phiZ98 [152], which belong to the dsRNA *Cystoviridae* family, utilize the lipopolysaccharide (LPS) core oligosaccharide of *P. aeruginosa* as the binding receptor. Smooth-colony-type *P. aeruginosa* strains exhibit resistance to phiYY due to the presence of the *galU* gene, which confers O-antigen to the LPS core oligosaccharide [145]. However, LPS-deficient *P. aeruginosa*, a strain that emerged during dsDNA phage therapy and lacks the *galU* gene [153], exposes its LPS core oligosaccharide on the cell surface, making it susceptible to phiYY [145]. Phage therapy employing phiYY has been reported to reduce bacterial load and alleviate clinical symptoms in patients diagnosed with interstitial lung disease (ILD) and chronic lung infections [146]. Moreover, a phage cocktail, including phiYY among five phages, has demonstrated effectiveness against a broad spectrum of *P. aeruginosa* clinical isolates and significantly impedes the emergence of phage-resistant mutants [145].

dsRNA phages have also been effectively applied as plant therapeutics. *Pseudomonas* phage phi6 has been employed to manage various plant diseases, including controlling *P. syringae pv. phaseolicola* (Pph), the causal agent of halo disease in common bean (*Phaseolus vulgaris*) [154]. Additionally, phi6 has been utilized against *P. syringae pv. actinidiae* (Psa), responsible for kiwifruit psyllid [155], and *P. syringae pv. syringae* (Pss), which causes early leaf symptoms of bean brown spot [154]. Other *Pseudomonas* phages, including phi8, phi12, phi13, phi2954, phiNN, and phiYY, have also shown efficacy against Pph [156]. Collectively, the application of dsRNA phages holds significant promise for complementing and enhancing phage therapy for both human and plant diseases.

### 4.3. Engineering ssDNA and RNA Phages for Antimicrobial Therapy

RNA phages exhibit significant potential as antigen carriers in the development of vaccines targeting drug-resistant pathogens. Huo and colleagues pioneered the development of Qβ phage presenting Qβ-glycan 1, a synthetic tetrasaccharide from *Salmonella enteritidis*. This construct successfully induced robust IgG antibody responses in both mice and rabbits [147]. Notably, carbohydrate-based antigens, such as tetrasaccharide, do not directly interact with helper T cells, leading to limited immune memory. However, the conjugation of glycan antigens to a potent carrier such as Qβ enables engagement with both T-helper and B cells, resulting in a vigorous antibody response. Mice treated with post-immunization serum recovered from rabbits vaccinated with Qβ-glycan 1 demonstrated increased survival rates following administration of lethal doses of *S. enteritidis*. In another study, Rashidijahanabad and colleagues reported the production of Qβ-OSP, a Qβ phage presenting O-specific polysaccharide (OSP) 1 of *Vibrio cholerae*. Employed in mouse immunization, the Qβ-OSP conjugate elicited strong IgG antibody responses against *V. cholerae* O1 Inaba, reaching sufficient IgG antibody levels after just two administrations. Remarkably, the titer of IgG antibodies remained detectable up to day 265 [148].

Phagemid vectors derived from the filamentous ssDNA M13 phage serve as a highly efficient engineering platform for diverse therapeutic applications, including engineering phages for enhanced antibacterial activity. The delivery of antibacterial therapeutics can be achieved either through surface display on the virion or incorporated into the M13 phagemid itself. These displayed therapeutics may target vital cell components, virulent factors, or entire bacterial pathogens. Notably, several potential candidates have been identified for combating *S. aureus*, *P. aeruginosa*, *H. pylori*, and other pathogens. In another example, the dual display of two functional peptides enables M13 to (1) undergo endocytosis by eukaryotic cells and (2) impede infection caused by the intracellular pathogen *Chlamydia trachomatis* [157]. For an in-depth exploration of anti-infective development using phage display, refer to [158].

Antibacterial efficacy can also be achieved through cargoes loaded onto M13 phagemid. Lu and Collins cleverly designed an indirect antibacterial method using M13 phages overexpressing *lexA3*, a repressor of the SOS DNA repair system [149]. The authors demonstrated that suppressing the SOS network in *E. coli* with engineered *lexA3*-M13 significantly enhanced quinolone killing and increased the survival of infected mice. The engineered phage holds promise for targeting antibiotic-resistant bacteria and biofilm cells and modulating the bacterial population resistant to antibiotics post-treatment. In a parallel approach, Prokopczuk and colleagues employed a filamentous phage Pf for an indirect strategy to interfere with *P. aeruginosa* pathogenesis at burn wound sites [159]. Given that *P. aeruginosa* is a major cause of burn-related infections and sepsis, often exhibiting multi-drug resistance, the authors engineered a superinfective Pf phage (eSI-Pf). Administration of eSI-Pf effectively attenuated *P. aeruginosa* virulence, reduced bacterial load at the wound site, minimized bacterial dissemination from the burn site to internal organs, alleviated septicemia symptoms, and ultimately improved mouse survival.

Direct bactericidal effects have been achieved through several successful therapies, including the delivery of programmed CRISPR/Cas by recombinant M13 phages. In a proof-of-concept study on phage therapy for strain-specific depletion and genomic deletions in the gut microbiome, Lam and colleagues engineered M13 to deliver *gfp*-targeting CRISPR/Cas9 to bacteria residing in the gastrointestinal tract [160]. The engineered phage showed efficient sequence-specific targeting of GFP-expressing *E. coli* in the gut, confirming the capability of CRISPR/Cas9 to induce genomic deletions at the designated target site.

In our laboratory, utilizing M13 phagemids, we spearheaded the development of a diverse range of CRISPR/Cas13a-based, sequence-specific antibacterial nucleocapsids, referred to as CapsidCas13a(s) [65]. Our research showcased the versatility of M13-based CapsidCas13a(s) in applications such as (1) bacterial gene detection, (2) modification of bacterial flora, and (3) therapeutics for AMR bacterial infections. In the first instance, we developed M13-based EC-CapsidM13Cas13a::KanR(s) for the detection of different genotypes of carbapenem resistance genes (*bla*_IMP-1_, *bla*_OXA-48_, and *bla*_VIM-2_). In the presence of corresponding target genes, CRISPR/Cas13a will be activated, and the subsequent non-specific RNase activity will result in host cell death. We observed specific activity of CapsidCas13a against target genes on both plasmids and chromosomes. In the second example, we demonstrated the potential of CapsidCas13a as a tool for modifying bacterial flora. Individual treatment of nontargeting, *bla*_IMP-1_-, or *mcr-2*-targeting CapsidCas13a with a mixed cell population of *E. coli* NEB5α F′I^q^ (control) and NEB5α F′I^q^ expressing *bla*_IMP-1_ and *mcr-2* at equal numbers resulted in a reduction in the corresponding target cell population from over 34% to less than 4%. In the last example, we examined the therapeutic effect of EC-CapsidCas13a-*bla*_IMP-1_ using a *Galleria mellonella* infection model. Administration of EC-CapsidCas13a-*bla*_IMP-1_ at MOI 100 to *G. mellonella* larvae infected with R10-61 (carbapenem-resistant clinical isolates of *E. coli* carrying *bla*_IMP-1_) significantly improved host survival compared to controls, EC-CapsidCas13a-nontargeting, and phosphate-buffered saline (PBS). Our work underscores the feasibility of harnessing phages as antibacterial therapeutics for diverse applications to control AMR bacterial infections and bacterial flora.

## 5. Concluding Thoughts and Future Outlook

Despite enormous advancements in biotechnology in the last century, nature’s true diversity and richness remain largely unexplored. Regarding ssDNA and RNA phages, their uncharted biodiversity is a rich resource awaiting to be uncovered. With the help of precision gene editing technology and synthetic biology, modern phage therapy is distinct from the once-forgotten conventional phage therapy in their usage of rationally designed engineered phages with enhanced properties. Engineered phages, benefiting from their simplicity and compatibility with prokaryotic hosts, offer a more cost-effective and faster production alternative compared to various systems, including eukaryotic viral ones.

The small, simple genomes of most RNA and ssDNA phages are amenable to genetic engineering, presenting the potential to overcome transformation barriers. This simplicity proves advantageous in synthetic engineering, with cell-free transcription-coupled translation (TX-TL) expected to operate efficiently. Despite these promising aspects, ssDNA and RNA phages face technical hurdles limiting their use in antimicrobial therapy. A significant challenge lies in the absence of an efficient isolation and culture method, stemming from our limited understanding of their natural hosts. The non-lytic chronic infection life cycle of certain ssDNA and RNA phages also poses challenges in developing isolation assays, hindering the use of conventional plaque assays. Additionally, computational approaches are needed to detect virus sequences in whole-genome shotgun sequencing data due to the low number of known shared genes between viruses.

On the positive side, ssDNA and RNA phages possess distinct features that make them valuable as antibacterial agents, particularly in combination therapy with dsDNA phages. Their unique host range, employing distinct receptors for host binding and entry, sets them apart. Their genetic nature, ssDNA or RNA, renders them immune to many dsDNA-targeting host defenses. Furthermore, they are likely to engage different host factors than dsDNA phages during their infection cycle, paving the way for therapeutic cocktails that simultaneously target multiple vital host components.

The untapped biodiversity of ssDNA and RNA phages, coupled with their simplicity and genetic amenability, positions them as valuable resources for future biotechnological and medical developments. Their unique features, including distinct host receptors and immunity to certain host defenses, open avenues for tailored therapeutic cocktails and innovative engineering strategies. In navigating this uncharted territory, the impact of RNA and ssDNA phages is poised to expand, offering solutions to current and emerging challenges. In this review, we underscore the importance of exploring and harnessing the diversity of nature, providing inspiration for future endeavors in synthetic biology and phage therapy. With ongoing advancements in precision gene editing and synthetic biology, the potential applications of these phages are limited only by our imagination, and the journey of exploration continues to unfold.

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
