# Peer review of "RNA and Single-Stranded DNA Phages: Unveiling the Promise from the Underexplored World of Viruses"

_ijms, 2023, doi:10.3390/ijms242317029_

Round 1

Reviewer 1 Report

Comments and Suggestions for Authors

General Comments

In the manuscript, the authors show a good review about the potential of RNA and ssDNA phages in the biotechnology and medicine. They explore the past and present applications of this underexplored group of phages, emphasizing its use as therapeutic agents, CRISPR/CAS-based genome edition, disease treatment, imaging, biomaterial development and delivery systems. The manuscript is easily read and is well supported with many references, although some of them need to be revised. The tables presented in the document are very useful and summarize the content of the work very well. The figures are quite schematic too.

Specific comments

There is a mistake in the line numbers and it start again after table 1 (pag 4). For this reason, I write the line and page number for each comment.

Line 85-87 (pag 2): “Except a few well-studied ssDNA phages, such as M13 or phiX174, functions of the remaining encoded proteins remain unknown [23], [24] [25][26]. Ref 23 and 24 contain information about Polyomaviridae (which are dsDNA viruses) and Flaviviridae (included viruses that infect mammals and arthropod) family respectively. Could you change it for a more appropriate one?

Line 99-103 (pag 3): “Interestingly, these three families infect completely different types of hosts. 99 Inoviridae infect Gram-negative bacteria with a lipopolysaccharide (LPS)-containing outer membrane, while Paulinoviridae only infect hosts lacking LPS, including both Gram-negative and Gram-positive hosts. Plectroviridae infect cell wall-less bacteria, such as members of the Mollicutes order [27].” The reference 27 only shows part of the information described in the paragraph. This does not show information about LPS. The exception is the supplementary figure 7, where mentions simple diderm (no LPS), but this category is not assigned a none virus group. Could the authors add another reference to support this information (“Paulinoviridae only infect hosts lacking LPS”).

Table 1: For the reference 27, the same comments than lines 99-103. I suggest including another reference to explain the LPS- Gram-negative bacteria, maybe this one: doi:10.1007/s10482-011-9616-8.

Table 1: The reference 34 was not found. The authors of this reference are associated to another virus group, Asfarviridae. Could the authors change it for another more appropriate one?

Table 1: I cannot found the reference 35, is it correct? I found works where cited a similar reference with an extra author “Davis J, Strauss J, Sinsheimer R”

Table 1: “Duinvirididae” should be change by “Duinviridae

Line 7 and 9 (pag 5): The ref 34 could be a mistake, the authors of this reference are associated to Asfarviridae, not Cystoviridae.

Line 11-12 (pag 5): “phages were subsequently isolated into two groups: those closely related (phi13) and those distantly related (phi8 and phi12) to phi6.” According to reference 39 this sentence is not completely true. The closest relatives of phi6 are phi7, phi9, phi10 and phi11. It appears that phi8 and phi12 are quite distant genetically from phi6 and the closely related phages, while phi13 is between the two groups.

Line 12-14 (pag 5): “Phi6 and its closely related relatives utilize type IV pili as the host receptor, limiting their infection range to P. syringae and several mutants of P. pseudoalcaligenes ERA [38]” I suggest replace the reference 38 for 39 in this statement.

Line 15-16 (pag 5): “Phi12 could infect other rough-LPS-containing Gram-negative bacteria such as E. coli JM109, though plaque formation was not observed [39]” According to ref 39, phi 8 and phi13 has the same effect. Why do the authors only mention phi12 in this sentence?

Lines 25-26 (pag 5): “No higher taxonomical order was assigned to this family”. In the ICTV master list 2019v1 appears orden Levivirales. Could the authors change this statement?

Lines 27-28 (pag 5): “Caulobacter phage phiCB5 and Pseudomonas phage PRR1, remained unclassified members that do not belong to any genera”. These viruses are classified and appear as: Cebevirus halophobicum (Taxonomy browser (Caulobacter phage phiCb5) (nih.gov)) and Perrunavirus olsenii (Taxonomy browser (Pseudomonas phage PRR1) (nih.gov)).

Lines 32-34 (pag 5): “although instead of binding to the tip of the pilus, both dsRNA and ssRNA phages bind to the side of the pilus to start their infection cycle [44].” In this reference, the lateral binding to the pilus is not mention. Could the authors propose another reference?

Lines 44-45 (pag 5): “Leviviricetes class, encompassing two orders (Norzivirales and Timlovirales) with six families, 428 genera, and 882 species”. The authors could cite the new ICTV master list for this statement.

Lines 61-63 (pag 6): “The authors recommended recovering viral samples at a lower buoyant density, and preparing the metagenomic library using multiple displacement amplification method to ensure proper enrichment of ssDNA samples.” According to reference 22, the authors say: “The continuous improvement of sequencing techniques including the development of library preparation kits that use ssDNA as starting material (Swift Biosciences) promise to overcome the community distortion resulting from multiple displacement amplification”. Then, I think the idea is: prepare libraries using ssDNA as starting material to ensure proper enrichment of ssDNA samples because the multiple displacement amplification method is a bias inducer. Could the authors check the reference?

Lines 95-98 (pag 6): “Our group, among others, has pioneered the development of phage-based RNA-cleaving antibacterials, encompassing sequence-specific bacterial gene detection, selective elimination of drug-resistant bacteria, and targeted manipulation of the bacterial flora [63]” This reference does not correspond to any author of this manuscript. I think the reference cited should be 148.

Table 2: Reference 69 does not show information about cancer treatment.

Table 2: Reference 73 mentions phages in general way in several sections and when electrospinning is named, dsDNA phages only appear associated to this content. Could the authors propose another reference?

Lines 122-125 (pag 8): “The engineered M13 phages could target and label abnormal collagens which were then easily detected by fluorescence imaging, enabling monitoring and diagnosis of various pathological conditions [69].” This reference does not mention any collagen. Could the authors add the adequate reference?

Lines 137-140 (pag 8): “These nanofibers can be used as efficient drug delivery systems for various therapies, incorporating bioactive molecules like antibiotics, anti-inflammatories, anti-cancer drugs, genomic DNA, proteins, probiotics, and enzymes [72], [73].” The reference 73 only mention dsDNA phages for this application. Could the authors add a reference where ssDNA o RNA phages are used?

Lines 148-151 (pag 8): “Dong and colleagues used silver nanoparticle-conjugated M13 phages that caused deconstruction of bacterial cell walls and intracellular biomolecules, inducing oxidative stress that killed pro-tumoral bacteria for gut microbiota regulation [70], [106].” The reference 106 is about nanoparticles, and the M13 phage is not mention. The authors could remove or change it.

Lines 215-216 (pag 9): “Recombinant VLPs of ssRNA phages can be generated by expressing the CP gene in bacteria or yeast [113], [114][127].” The reference 113 does not mention VLP, and the other references show information in bacteria system, not in yeast. Could the authors check these references to correct this sentence?

Lines 269-272 (pag 11): “Co-expression of phage coat proteins and suitable RNA templates from transformed plasmids. Recombinant VLPs can be assembled through the direct interaction between coat proteins and RNA molecules [123]” Could the authors suggest another reference? In the ref 123 does not explicitly mention how the VLPs are generated from the recombinant plasmids.

Lines 275-276 (pag 11): “after which genome reassortment can lead to various combinations of packaged genomic segments (step â‘£) [125].” In this reference does not show information about the packaging of segments from different source. It only describes how reporter genes could be lost by recombination. Could you suggest an adequate reference?

Lines 344-345 (pag 13): “or inhibition of cell wall synthesis using single-gene lysis (Slg) protein from ssDNA and ssRNA phages [139]–[142].” I understand that the authors are including the reference 141, however this one describes the phiYY lytic enzyme, which is a dsRNA phage. Could the authors modify this cite?, maybe writing [139], [142]. 

Lines 357-359 (pag 13): “Notably, recent genome analyses of the animal dsRNA viruses from Picobirnaviridae and Partitiviridae families have unveiled potential antibacterial and antifungal  single-gene lysis (Slg) genes [50].” Could the authors check this reference? I understood that in Picobirnavirus have been found lysis proteins, not slg genes; and CRISPR spacers have been detected in Partitiviruses.

Lines 359-361 (pag 13): “Upon cloning and expression of the Sgl genes from Picobirnavirus and Partitivirus in E. coli DH5a, the observed growth inhibition was comparable to that of induced by the Enterobacteria MS2 phage lysin L [149].”  This reference show information about Picobirnavirus bacteriolytic protein, not slg gene. Could the authors modify this sentence?

Lines 365-367 (pag 13): “On other hand, SglQβ from Qβ phage exhibits bacteriolytic activity by non-competitively inhibiting MurA, the first enzyme in the PG biosynthetic pathway [139], [140].” In the reference 139, there is not mention that SglQβ has a non-competitively inhibition on MurA. In this reference describe this kind inhibition for E protein from phiX174 on MraY. Could the authors check this reference?

Lines 378-379 (pag 14): “Pseudomonas phages phiYY [42] and phiZ98 [150], belong to the dsRNA Cystoviridae family” The reference 150 does not mention the phage phiZ98. Could the authors change it?

Lines 380-381 (pag 14): “Smooth-colony-type P. aeruginosa strains exhibit resistant to phiYY due to the presence of the galU gene, which confers O-antigen to the LPS core oligosaccharide” Please add a reference for this statement, maybe the reference 143 could be adequate.

Lines 390-393 (pag 14): “Pseudomonas phage phi6 has been employed to manage various plant diseases, including controlling P. syringae pv. phaseolicola (Pph), the causal agent of halo disease in common bean (Phaseolus vulgaris) [151].” The reference 151 does not show treatment based on phage phi6. Could the authors provide an adequate reference?

Lines 395-396 (pag 14): “Other Pseudomonas phages, including phi8, phi12, phi13, phi2954, phiNN, and phiYY, have also shown efficacy against Psa [153].” This reference does not show experiments using these phages. Could the authors provide an adequate reference?

Lines 441-444 (pag15): “In a proof-of-concept study on phage therapy for strain-specific depletion and genomic deletions in the gut microbiome, Lam and colleagues engineered M13 to deliver gft-targeting CRISPR/Cas9 to bacteria residing in the gastrointestinal tract [157]” I did not find the word gft-targeting in this reference. Maybe the authors would say GFP-targeting.

Reviewer 2 Report

Comments and Suggestions for Authors

The manuscript of Huong Minh Nguyen et al. describes possible applications of RNA and dsDNA bacteriophages in biotechnology and biomedicine. The manuscript contains information of interest to researchers, both about general group characteristics and particular information regarding the interaction of the phage with the host and host receptors. The manuscript is well organised and is written in good and comprehensive language, the figures and tables are nice and helpful, but  a few comments can be made..The reviewer’s general concern is related to the lack of comparisons of the effectiveness  of tailed dsDNA phages and ssDNA and RNA phages for phage therapy/control of human and plant diseases.

Probably, it would make sense to modify the title of the manuscript in order to show that it covers more than only therapeutic topics.

Line 42 - It would be better to replace the word “known” with the word “tailed”. You forgot about Tectiviridae phages, technically they are also dsDNA phages. It is also worth noting that a decent number of families, subfamilies and genera are not classified as higher-ranking taxa, with the exception of the class Caudoviricetes.

Lines 62-63 - It would be interesting to list those multiple distinct features which make ssDNA and RNA phages an attractive addition to dsDNA phages.

Lines 83-84 - Please check genome size and number of ORFs of sequences  filamentous phages (for example Vibrio phage K05K4_VK05K4 #CP017905  21,012 bp 34 CDSs).

Line 87 - check the formatting of references.
